

# SMRT sequencing of the full-length transcriptome of the white-backed planthopper *Sogatella furcifera*

Jing Chen[1], Yaya Yu[1], Kui Kang[2] and Daowei Zhang[2]

[1] College of Basic Medical Science, Zunyi Medical University, Zunyi, China
[2] School of Biological and Agricultural Science and Technology, Zunyi Normal University, Zunyi, China

## ABSTRACT

The white-backed planthopper *Sogatella furcifera* is an economically important rice pest distributed throughout Asia. It damages rice crops by sucking phloem sap, resulting in stunted growth and plant virus transmission. We aimed to obtain the full-length transcriptome data of *S. furcifera* using PacBio single-molecule real-time (SMRT) sequencing. Total RNA extracted from *S. furcifera* at various developmental stages (egg, larval, and adult stages) was mixed and used to generate a full-length transcriptome for SMRT sequencing. Long non-coding RNA (lncRNA) identification, full-length coding sequence prediction, full-length non-chimeric (FLNC) read detection, simple sequence repeat (SSR) analysis, transcription factor detection, and transcript functional annotation were performed. A total of 12,514,449 subreads (15.64 Gbp, clean reads) were generated, including 630,447 circular consensus sequences and 388,348 FLNC reads. Transcript cluster analysis of the FLNC reads revealed 251,109 consensus reads including 29,700 high-quality reads. Additionally, 100,360 SSRs and 121,395 coding sequences were identified using SSR analysis and ANGEL software, respectively. Furthermore, 44,324 lncRNAs were annotated using four tools and 1,288 transcription factors were identified. In total, 95,495 transcripts were functionally annotated based on searches of seven different databases. To the best of our knowledge, this is the first study of the full-length transcriptome of the white-backed planthopper obtained using SMRT sequencing. The acquired transcriptome data can facilitate further studies on the ecological and viral-host interactions of this agricultural pest.

Corresponding author
Daowei Zhang,
zhangdaowei@zync.edu.cn

## INTRODUCTION

The white-backed planthopper (WBPH) *Sogatella furcifera* (Hemiptera: Delphacidae) is an economically destructive rice pest distributed throughout Asia (*Ma et al., 2017*; *Matsumura et al., 2014*). As a migratory pest, *S. furcifera* has a great ability to spread, causing widespread damage to crops (*Wang et al., 2012*). *S. furcifera* mainly affects crops in two ways, by sucking phloem sap and causing growth delay (*Zhou et al., 2013*); in severe cases, it causes 'hopper burn' (*Lei et al., 2014*; *Wang et al., 2017*). Furthermore, *S. furcifera* transmits plant viruses such as southern rice black-streaked dwarf virus (SRBSDV), rice ragged stunt virus (RRSV), and rice grassy stunt virus (RGSV) (*Chen et al., 2018*; *He et al., 2016*; *Zhou et al., 2013*).

Currently, this pest is controlled by extensive application of chemical insecticides (*Jin et al., 2017*; *Matsumura et al., 2014*; *Yang et al., 2018*), but long-term use of chemical agents can lead to the development of pest resistance, causing re-infestation (*Matsumura et al., 2014*; *Zhou et al., 2013*). In addition, the overuse of pesticides can lead to environmental pollution and safety issues. Therefore, a practical approach to pest control is urgent instead of using pesticides. For example, pest control can be carried out by regulating key processes such as growth and development, immune defenses, and reproduction in pests. The target genes regulating these processes should be identified and isolated to develop new approaches to pest control. Therefore, the acquisition of long-fragment gene sequences is necessary for studies on gene functions and to identify genes that can be targeted for pest control applications.

The transcriptome can reflect the number and types of genes in a cell, revealing biochemical processes at the physiological and cellular levels (*Alba et al., 2005*). The next generation of high-throughput sequencing, also known as second-generation sequencing, has enabled a better understanding of various gene expression and regulation mechanisms (*Bao et al., 2013*). This method does not require a reference genome, and therefore it is suitable for non-model species (*Chao et al., 2018*). In 2017, genome and transcriptome analyses of *S. furcifera* were performed using this method (*Wang et al., 2017*). De novo assembly and characterization of the *S. furcifera* transcriptome provided comprehensive data for gene function identification. The RNA-Seq data served as a basis for screening and analyzing candidate genes in *S. furcifera* for pest control. As a cogent tool for understanding gene expression and identifying splice junctions, second-generation sequencing has become a focus of research; however, it does not provide the full-length sequence of all genes (*Djebali et al., 2012*; *Nagalakshmi et al., 2008*; *Zhang et al., 2018*). A full-length transcript is defined as the entire transcript from the 5′ to 3′ end and containing the poly-A tail. However, owing to the read-length limitation in second-generation sequencing, the transcript obtained by this method is incomplete (*Wang et al., 2017*). Single-molecule real-time (SMRT) sequencing, a third-generation sequencing technology developed by Pacific Biosciences (PacBio), can effectively overcome this problem (*Eid et al., 2009*; *Korlach et al., 2010*). With its long read-length advantage, it can directly read reverse-transcribed full-length complementary DNA (cDNA) and provide high-quality, long, and intact transcripts (*Sharon et al., 2013*). The average read length is 10–15 kb and the maximum length is up to 70 kb (*Stadermann, Weisshaar & Holtgrawe, 2015*), which helps the accurate identification of isoforms that share exons (*Faist et al., 2009*). Furthermore, this method can identify new genes and complement genome annotations, and contribute to the accurate analysis of fusion genes, homologous genes, and superfamily genes or alleles.

PacBio SMRT sequencing has been used for whole-transcriptome profiling of several plants, animals, and microorganisms and numerous high-confidence transcripts have been obtained (*Chao et al., 2018*; *Jia et al., 2018*; *Zeng et al., 2018*). Although the genome of *S. furcifera* has been partially sequenced (*Wang et al., 2017*), information about the structure and sequence of transcripts is limited and the genes have not been well annotated. In the present study, the full-length transcriptome of *S. furcifera* was obtained by SMRT sequencing. Using the procured transcriptome data, we performed full-length
sequence prediction, SSR analysis, lncRNA prediction, and gene function annotation. The transcriptome data provide numerous full-length sequences of *S. furcifera*, which will accelerate genome annotation and enhance our understanding of the genetic structure and diversity of *S. furcifera*.

## MATERIALS & METHODS

### Insects

Inbred laboratory strains of *S. furcifera* originated from Zunyi Normal University of China. Continuous generations of *S. furcifera* were maintained on the susceptible rice variety Taichung Native 1 (TN1) in an artificial climatic chamber under the following stable conditions: 27 ± 2 °C, 16:8 h light/dark photoperiod, and 75% ± 5% relative humidity for three years, spanning at least 30 generations. The eggs laid by *S. furcifera* within 24 h, first to six instar larvae, and newly emerged adults starved for 24 h were collected and washed in precooled normal saline. Finally, 0.4 g of eggs, 2.0 g of larvae, and 5.0 g of adults were harvested for RNA sample preparation.

### RNA sample preparation

Total RNA samples (at three developmental stages) were isolated using an RNA extraction kit (Tiangen Biotech Co., Ltd., Beijing, China). Contamination and degradation of RNA were assessed by electrophoresis on 2% agarose gels. RNA quantity was determined using a Nanodrop 2000c spectrophotometer (Thermo Fisher Scientific, Waltham, MA, USA). The concentration of RNA was assessed using the Qubit RNA IQ Assay Kit with the Qubit Fluorometer (Thermo Fisher Scientific). The integrity of RNA was determined using the 2100 Agilent Bioanalyzer system (Agilent Technologies, Santa Clara, CA, USA).

### Library construction

The purified RNA products were sent to Novogene, Beijing, China, for SMRTbell™ library preparation and sequencing according to the manufacturer's protocol for the PacBio Sequel System (Pacific Biosciences, Menlo Park, CA, USA). Firstly, First-strand cDNA was amplified from the total RNA samples using the SMARTer™ PCR cDNA Synthesis Kit (Takara Bio USA, Inc., Mountain View, CA, USA) according to the manufacturer's protocol, and then the quality of cDNA was assessed by OD 260/280 ratio and gel electrophoresis. Finally, the double stranded cDNA was amplified by polymerase chain reaction (PCR). The BluePippin automated DNA size selection system (Sage Science, Beverly, MA, USA) was utilized to further narrow the size distribution of the final libraries (n < 4 kb and n > 4 kb). Damage repair, end repair, and SMRT dumbbell joint attachment to full-length cDNA were carried out. Finally, sequencing was performed using the PacBio Sequel single-molecule real-time sequencer (Pacific Biosciences).

### Preprocessing of SMRT sequencing reads

The sequences were aligned using SMRT software (version 5.0) with the following parameters: minReadScore = 0.75 and minLength = 200. The quality of transcriptome completeness was assessed using Benchmarking Universal Single-Copy Orthologs (BUSCO)

version 3 (*Simao et al., 2015*) with the eukaryota database. Circular consensus sequences (CCSs) were obtained from subread BAM files (parameters: max_drop_fraction, 0.8; max_length, 18000; min_length, 200; min_predicted_accuracy, 0.8; min_passes, 1; min_zscore, -999; no_polish, TRUE), with the output in CCS.BAM file format. By searching for the 5′ and 3′ end-attached adapters and the poly(A) tail, large CCSs were clustered into full-length and non-full-length reads. The full-length reads contained both 5′ and 3′ adapters and poly(A) tails. The CCSs, including these three elements and no excess copies of attached adapters within the DNA sequence, were classified as FLNC. Consensus isoforms were then obtained using Iterative Clustering for Error Correction (ICE) clustering analysis with FLNC. In addition, the consensus isoforms (specifically refers to one of the many splice variants of a gene) were polished using the non-full-length reads to identify high-quality isoforms, using Quiver for more than 99% correction accuracy (hq_quiver_min_accuracy, 0.99; qv_trim_3p, 30; qv_trim_5p, 100; bin_size_kb, 1; bin_by_primer, false).

## Preprocessing of non-redundant transcripts

CD-HIT (*Fu et al., 2012*) was used to obtain non-redundant transcripts. It is clustering and redundancy software that removes redundant and similar sequences by sequence alignment clustering, with the final output in non-redundant (Nr) sequence file format. The corrected transcript sequences were non-redundant according to 99% similarity, and the distribution of length frequency before and after correction for non-redundancy was counted.

## CDS prediction

The ANGEL pipeline, which is based on ANGLE (*Shimizu, Adachi & Muraoka, 2006*), was used to identify the protein coding regions from cDNAs (https://github.com/PacificBiosciences/ANGEL). We then identified the protein sequences of *S. furcifera*, and closely related species were used for ANGEL training and prediction. Eventually, transcripts containing the 5′- and 3′-UTRs (untranslated regions) and complete CDSs were defined as full-length transcripts.

## Transcription factor (TF) analysis

AnimalTFDB (Animal Transcription Factor Database v2.0, (http://bioinfo.life.hust.edu.cn/AnimalTFDB2/)) (*Zhang et al., 2015*) was used to analyze TFs in the *S. furcifera* transcriptome database. Because *S. furcifera* is not included in the database, the TFs were identified using hmmsearch based on the Protein Family (Pfam) search results of the TF family.

## lncRNA analysis

Coding and non-coding transcripts were categorized using the following four coding potential analysis tools: Coding Potential Calculator (CPC) (*Kong et al., 2007*), Coding-Non-Coding Index (CNCI) (*Sun et al., 2013*), Predictor of long non-coding RNAs and messenger RNAs based on an improved k-mer scheme (Plek) (*Li, Zhang & Zhou, 2014*), and Pfam-scan (Pfam) (*Finn et al., 2016*). Transcripts less than 200 bp, predicted using these four tools, were removed. The CNCI tool can effectively differentiate the coding sequence

of proteins and non-coding sequences free of annotations of known genes. CPC, primarily by assessing the protein-coding potential of a transcript based on biologically meaningful sequence features and identifying sequences in protein sequence databases, classifies the transcripts into coding and non-coding sequences. The NCBI eukaryotes' protein database with an e-value of $e^{-10}$ was used for CPC analysis in our study. The transcript sequences predicted using Plek, CNCI, and CPC tools were used to search the Pfam-A and Pfam-B databases using hmmscan.The Pfam-A database contained high-quality domains of most known proteins, while the Pfam-B database contained more comprehensive domain families including a large number of small families for which few representative sequences are known. The transcripts that contained Pfam domains were eliminated in subsequent steps; default parameters were used for Pfam searches, -E 0.001 and –domE 0.001.

### SSR analysis

SSRs of the transcriptome were identified using MISA 1.0 (http://pgrc.ipk-gatersleben. de/misa/misa.html) (*Beier et al., 2017*). The unit sizes and their minimum number of repetitions were: 1-10, 2-6, 3-5, 4-5, 5-5, and 6-5. For example, 1-10 indicates that a single nucleotide is a repeating unit and the number of repetitions is at least 10 and 2-6 indicates that a dinucleotide is a repeating unit and the minimum number of repeats is 6.

### Gene function analysis

To obtain annotation information of unigenes, various databases were utilized. Transcripts were compared against the NCBI databases Non-redundant protein sequences (Nr, diamond v0.8.36) (*Li, Jaroszewski & Godzik, 2002*), NCBI non-redundant nucleotide sequences (Nt, ncbi-blast-2.7.1+), Swiss-Prot, Kyoto Encyclopedia of Genes and Genomes (KEGG, diamond v0.8.36) (*Kanehisa et al., 2004*) and Clusters of Orthologous Groups of proteins database (COG, diamond v0.8.36) (*Tatusov et al., 2003*) by BLASTX v2.2.31 with an $E$-value cut-off of $10^{-5}$. HMMER 3.1 was used to compare amino acid sequence transcripts against the Pfam database for Pfam annotation (*Ventsel, Kriuchkov & Nesterenko, 1973*). Gene Ontology (GO) were determined based on the protein annotation results of the Pfam database (*Ashburner et al., 2000*).

## RESULTS

### SMRT sequencing data output

The WBPH transcriptome was sequenced using the PacBio Sequel platform from a pooled RNA sample of *S. furcifera* obtained at different developmental stages, to accurately capture full-length transcript isoforms (BioSample accession: SAMN12612920). RNA was isolated from pooled samples and the cDNA was classified as fractions of full-length transcripts up to 4 kb (without performing size selection). The analysis of transcriptome completeness with BUSCO was shown in Table 1. A total of 15.64 Gbp sequence data in 12,514,449 PacBio subreads were obtained, with an average sequence length of 1,250 bp and N50 of 2,665 bp (Table 2, Fig. 1). To provide more accurate and reliable sequences, CCSs were generated from a single molecule sequence, and finally, 630,447 CCSs were obtained. By detecting the sequences, 421,026 CCSs were identified as full-length reads (containing 5′
**Table 1** **BUSCO analysis of assembly completeness.** The categories of genes are: (i) complete BUSCOs: genes which match genes in the BUSCO reference group. (ii) complete and single-copy BUSCOs: genes which match a single gene in the BUSCO reference group. (iii) complete and duplicated BUSCOs: if these are found more than once they are classified as 'duplicated'. (iv) fragmented BUSCOs: genes only partially recovered for which the gene length exceeds the alignment length cut-off. (v) missing BUSCOs: not recovered genes.

| BUSCO results | Count | Percentage (%) |
|---|---|---|
| complete BUSCOs | 958 | 70.08% |
| complete and single-copy BUSCOs | 605 | 44.26% |
| complete and duplicated BUSCOs | 353 | 25.82% |
| fragmented BUSCOs | 98 | 7.17% |
| missing BUSCOs | 311 | 22.75% |
| total | 1,367 | |

**Table 2** **Statistics of subread results.** (i) Subread base: size of the valid insert subreads. (ii) Subread number: number of valid insert subreads. (iii) Average subread length: average length of subreads. (vi) N50: represents the length of subreads that are more than 50% of the total length.

| Sample | Subread bases (G) | Subread number | Average subread length | N50 |
|---|---|---|---|---|
| *S. furcifera* | 15.64 | 12,514,449 | 1,250 | 2,665 |

and 3′ signals plus the poly(A) tails) and 388,348 were identified as FLNC reads with a mean length of 2,682 bp (Table 3, Fig. 1). The FLNC reads with highly similar sequences were clustered together into one consensus sequence using the ICE algorithm, resulting in 251,109 consensus isoforms (Table 3, Fig. 1), in combination with non-full-length sequences. Eventually, using the Quiver program to cluster non-full-length sequences, the resulting sequences were corrected, generating 29,700 high-quality isoforms (HQs).

## Non-redundant transcript acquisition

To obtain a series of non-redundant transcripts, we clustered highly similar coding sequences together using CD-HIT with an amino-acid sequence identity threshold of 99%. All 251,109 consensus transcripts were used for the acquisition of non-redundant transcripts, and finally, 156,138 unigenes (mean length of 2,994 bp) were obtained. The length distribution of the unigenes was counted at an interval of every 500–1,000 bp. As shown in Table 4, unigenes longer than 3,000 bp accounted for 46.6%, whereas those shorter than 500 bp only accounted for 8.9% of the total genes. As shown in Table 5, 92.5% of unigenes had only one isoform, while only 7.5% of unigenes had two to ten isoforms.

## SSR detection

A total of 100,360 SSR sequences (5,833,286 bp) were obtained, with 74,401 SSRs and 25,959 SSR-containing sequences. The cumulative number of sequences with at least one SSR was 73,671 and the number of SSRs present in compound formation was 27,268. Moreover, SSRs consisting of one to six (mono-, di-, tri-, tetra-, penta-, and hexa-nucleotide) tandem repeats were identified. Mono-repeats (46,035; 62.98%) were the most abundant in the WBPH unigenes, followed by tri-repeats (21,834; 29.87%) and di-repeats (4,453; 6.09%).
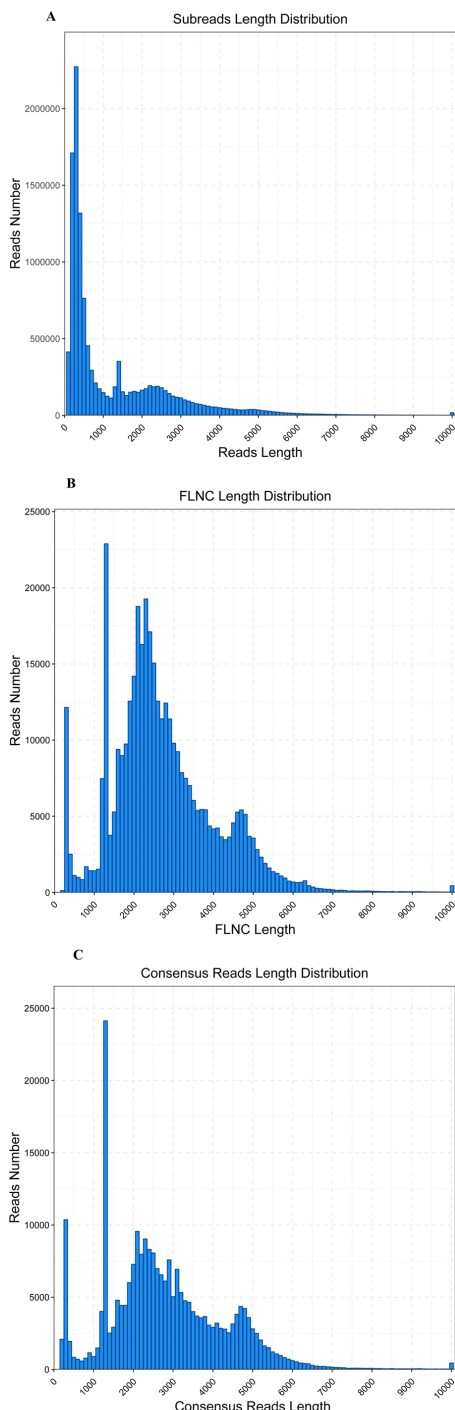

**Figure 1   Read length distribution of SMRT sequencing.** (A) Distribution of the number and length of 12,514,449 subread sequences. (B) Distribution of the number and length of 388,348 FLNC sequences. (C) Distribution of the number and length of 251,109 consensus isoforms.

**Table 3  Statistics of CCS results.** (i) CCS: circular consensus sequence. (ii) 5′-primer: number of reads with 5′signals. (iii) 3′- primer: number of reads with 3′signals. (iv) Poly-A: number of reads with Poly-A tails. (v) Full length: full-length reads. (vi) FLNC: full-length non-chimeric reads. (vii) Average FLNC read length: average length of full-length non-chimeric reads. (viii) Consensus reads: number of reads of the non-redundant sequence obtained after clustering.

| Sample | CCS | 5′-primer | 3′-primer | Poly-A | Full length | FLNC | Average FLNC read length | Consensus reads |
|--------|-----|-----------|-----------|--------|-------------|------|--------------------------|-----------------|
| *S. furcifera* | 630,447 | 547,929 | 562,407 | 503,155 | 421,026 | 388,348 | 2,682 | 251,109 |

**Table 4  Summary of the transcriptome assembly.** Percent of transcripts or unigenes (%): percentage of transcripts or unigenes in corresponding length range (bp).

| Length range (bp) | Transcripts | Unigenes |
|-------------------|-------------|----------|
| <500 | 15253 (6.1%) | 13966 (8.9%) |
| 500–1000 | 4139 (1.7%) | 2304 (1.5%) |
| 1000–2000 | 62045 (24.7%) | 31395 (20.1%) |
| 2000–3000 | 75229 (30.0%) | 35699 (22.9%) |
| >3000 | 94443 (37.5%) | 72774 (46.6%) |
| Total | 251109 | 156138 |

**Table 5  Number of genes corresponding to the transcripts.** 1/2/3/4/5/6/7/8/9/10: number of genes containing the same number of transcripts.

| Isoform number | 1 | 2 | 3 | 4 | 5 | 6 | 7 | 8 | 9 | 10 |
|----------------|---|---|---|---|---|---|---|---|---|----|
| Unigene number | 144,413 | 5,814 | 2,123 | 1,074 | 647 | 443 | 291 | 221 | 182 | 930 |

The frequencies of tetra-, penta-, and hexa-repeat types were only 0.86% (632), 0.14% (99), and 0.05% (39) in the WBPH unigenes, respectively (Fig. 2).

## TF detection

We analyzed TFs by comparing the transcript sequences to those in the AnimalTFDB 2.0 database, resulting in 1,288 TFs. The numbers of TFs enriched were as follows: zf-C2H2 (425), ZBTB (166), TF_bZIP (71), Homeobox (68), and HMG (60) (Fig. 3).

## Prediction of coding sequences

We used verified protein sequences of this species or closely related species for ANGEL training and then ran the ANGEL prediction for the given sequences. Then, 121,395 coding sequences were generated; among them, 48,873 coding sequences containing the start and stop codons were defined as complete open reading frames (ORFs). Subsequently, the number, size, and length distributions of the 5′ and 3′ UTR regions were analyzed. In total, 8,225 transcripts were annotated to the 5′-UTR regions and 2,231 to the 3′-UTR regions. As shown in Fig. 4, CDS lengths ≤ 1,000 bp accounted for 85.5% (103,696), followed by those from 1,000–2,000 bp (14,625; 12.0%) and >2000 bp (3,074; 2.5%).

## Identification of lncRNAs

We used four tools to identity unique transcripts without protein coding potential (i.e., lncRNAs). The CNCI tool identified 110,664 lncRNAs, Pfam identified 121,199 lncRNAs,

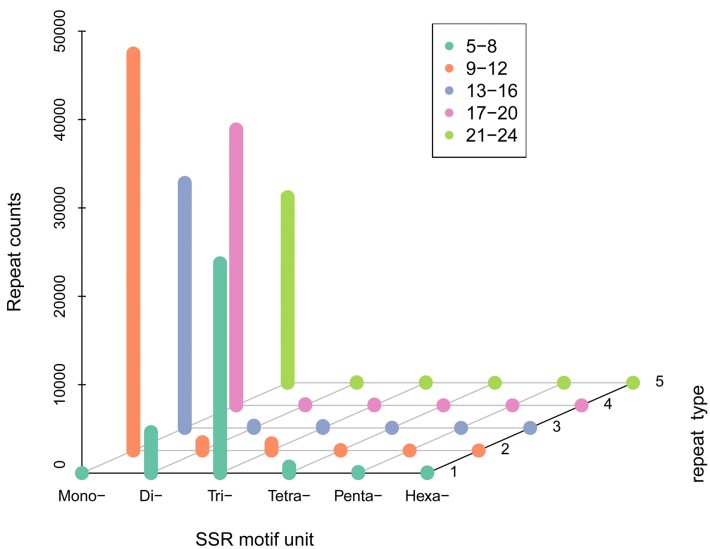

**Figure 2** **SSR density of different types of SSRs.** (i) SSR motif unit: number of repeating bases. (ii) "Mono-": repeat unit of a single base. (iii) "Di-": two bases. (iv) "Tri-": three bases. (v) "Tetra-": four bases. (vi) "Penta-": five bases and "Hexa-": six bases. The specific number of repetitions should correspond to the legend according to the color.

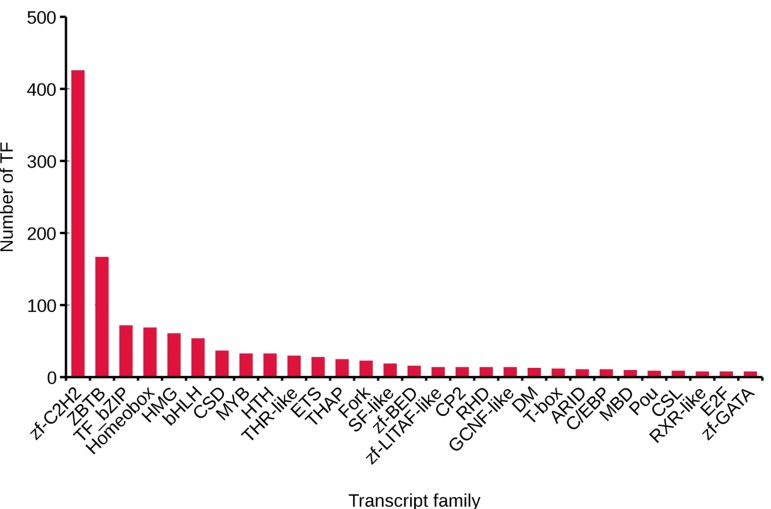

**Figure 3** **Transcript family distribution of TFs.** Different types of transcript family were plotted along the *x*-axis, while the number of transcription factors were plotted along the *y*-axis.

Plek identified 51,434 lncRNAs, and CPC identified 98,911 lncRNAs. In total, 44,324 lncRNA transcripts were predicted by all four methods (Fig. 5).

## Functional annotation of transcripts

All 156,138 corrected transcripts were utilized to annotate according to function via searching the GO, KEGG, COG, NR, NT, Pfam, and SwissProt databases, and 95,495

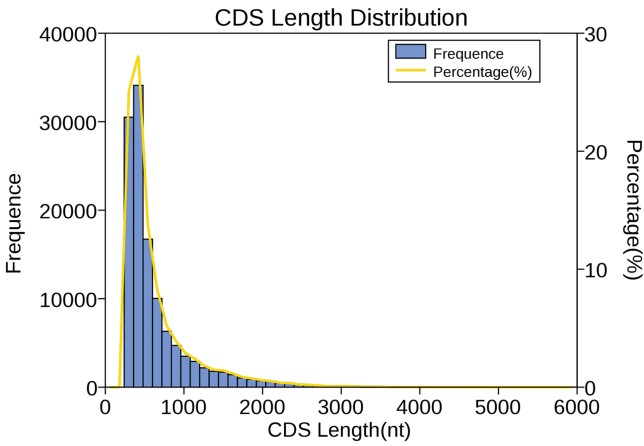

**Figure 4** **Length distribution of CDSs.** Length of predicted CDS was plotted along the *x*-axis, while number of CDS transcripts was plotted along the left *y*-axis. The yellow line that represents the percentage of CDS length was plotted along the right *y*-axis.

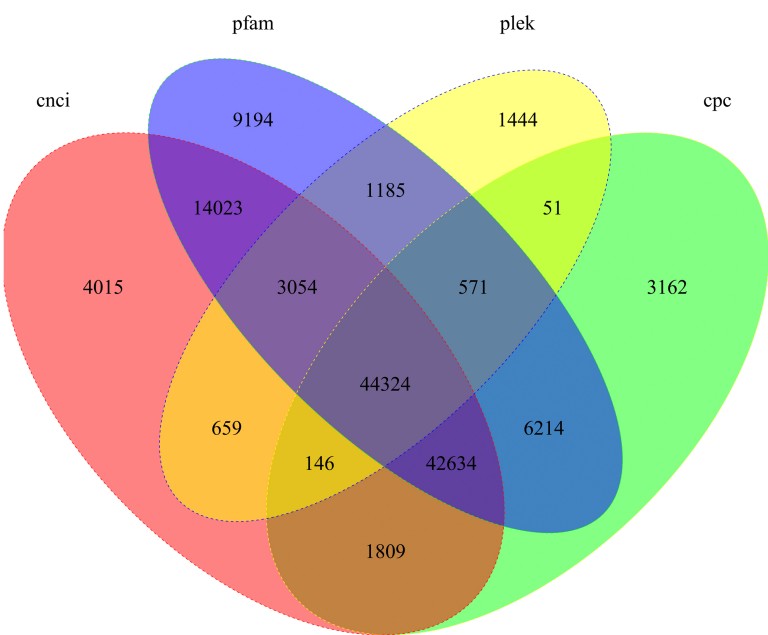

**Figure 5** **Candidate lncRNAs identified using CNCI, Pfam, Plek, and CPC.** Non-overlapping areas indicate the number of lncRNAs identified by a single tool. Overlapping areas indicate the total number of lncRNAs identified by several tools.

transcripts (61.2%) were annotated (Fig. 6A). Firstly, the transcripts were compared to those in the NR database (70,969 transcripts). The results showed that species with the most matching transcripts belong to Hemiptera insects including *Clastoptera arizonana* (11,731), *Nilaparvata lugens* (6,983), *Cuerna arida* (6,190), *Graphocephala atropunctata* (6,052), and *Homalodisca liturata* (5,485) (Fig. 6B). KEGG analysis showed that 60,352 transcripts were

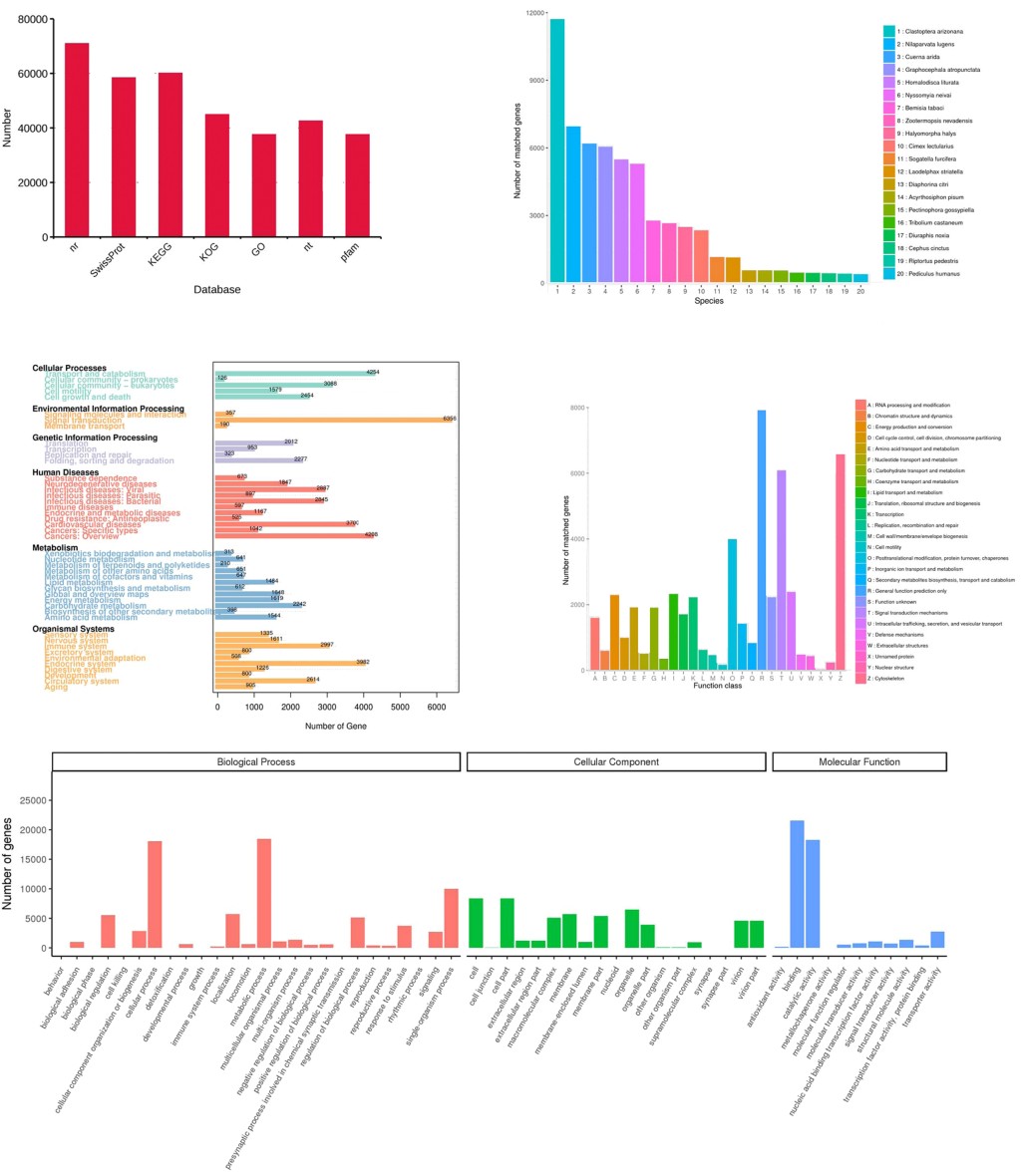

**Figure 6** **Functional annotation of the corrected isoforms.** (A) Function annotation of transcripts in all databases. GO, Gene Ontology; KEGG, Kyoto Encyclopedia of Genes and Genomes; Nr, Non-Redundant Protein Database; COG, Cluster of Orthologous Groups of proteins. (B) Species of highest scoring blastp match in Nr. (C) KEGG pathway assignment of transcripts. (D) COG annotation of transcript sequences. (E) Distribution of GO terms for all annotated transcripts in cellular component, biological process and molecular function.

assigned to 356 KEGG pathways and clustered significantly in signal transduction, transport and catabolism, and endocrine system (Fig. 6C). The COGs functional classification of the transcripts is shown in Fig. 6D, 44,948 transcripts matched an entry in COG. Finally, the result of GO analysis showed that 37,604 transcripts were enriched in the pathways related to biological processes, cellular components, and molecular functions. Most of the

genes were involved in the following "biological processes": cellular process, metabolic process, single-organism process, location, regulation of biological processes, and biological regulation. A high number of genes in "cellular components" were mainly involved in cell, cell part, membrane, organelle, and virion. The category "molecular functions" mainly consisted of transcripts involved in catalytic, binding, transporter, nucleic acid binding transcription factor, and structural molecular activities (Fig. 6E).

## DISCUSSION

The SMRT sequencing platform produces long reads that can effectively resolve the issue of completed gene structures. The strengths of PacBio RNA sequencing have been widely investigated in various species (Allen et al., 2017; Chao et al., 2018; Jia et al., 2018; Park et al., 2017; Sharon et al., 2013; Tombacz et al., 2017; Zeng et al., 2018). To date, a draft genome sequence of *S. furcifera* has been published (Wang et al., 2017), but the sequences of most full-length transcripts are still unknown. We mapped SMRT sequencing transcripts to the *S. furcifera* reference genome (Wang et al., 2017) using GMAP (Wu & Watanabe, 2005). The percentage of mapped transcripts occupied 64.79% and unmapped occupied 35.21% (Table S1). As we mentioned, rice virus SRBSDV, RRSV and RGSV were transmitted by *S. furcifera*, so there were some of the transcript sequences appear to be of viral origin in our sequencing results.

One of the advantages of SMRT sequencing is that it provides new comprehension of full-length sequences, gene structures, and gene functions. The results of our study indicate that SMRT sequencing is useful for both genome annotation and gene function studies (Dong et al., 2015). The full-length transcriptome of *S. furcifera* was analyzed using PacBio SMRT sequencing and 15.64 Gbp of clean data was obtained, including 630,447 CCSs, of which 388,348 were identified as FLNC transcripts. In addition, 251,109 consensus transcripts were obtained using transcript cluster analysis of the FLNC reads, which included 29,700 high-quality transcripts. The lengths of the FLNC transcripts were aligned to the sizes of the transcriptome library (Fig. 1B). PacBio CCSs and FLNC reads did not require assembly of short next generation sequencing (NGS) reads. Furthermore, the high competence of PacBio SMRT sequencing to generate full-length transcript sequence may well be connected with its long-read property. In our study, 156,138 unigenes with mean length of 2,994 bp were obtained from the SMRT sequencing platform, which is much better than those assembled using only Illumina sequencing. For example, Li et al. (2016) obtained 51,842 unigenes with a mean length of 746 bp, and Xu et al. (2012) obtained 81,388 unigenes with a mean length of 555 bp.

lncRNAs are a class of non-protein-encoding transcripts of over 200 nt in length that are important to regulate gene expression at various levels. Currently, the roles of lncRNAs in insects, such as *Drosophila melanogaster*, *Agasicles hygrophila*, *Plutella xylostella*, and *N. lugens*, have been reported. Studies on *D. melanogaster* have shown that lncRNAs are involved in sex determination (Mulvey et al., 2014), motor behavior and climbing ability (Li et al., 2012), courtship behavior in males (Chen et al., 2011), sleep behavior (Soshnev et al., 2011), and inactivation of X staining (Deng & Meller, 2006;

*Smith, Allis & Lucchesi, 2001*). In recent years, with increasingly intense research on lncRNA functions, the RNA-Seq data of some insects are being re-explored and several insect lncRNAs have been discovered. While studies on the function of lncRNA in fruit flies and honeybees have made crucial progress, in other insects, such as agricultural pests, the study of lncRNA function is impeded by several difficulties and challenges. In our study, 44,324 lncRNAs were predicted by all four methods, and their function in S. furcifera requires further investigation.

Full-length transcript sequence information is crucial for genome annotation and gene function research. However, most methods used to obtain a full-length cDNA clone are time consuming, expensive, and inefficient (*Bower & Johnston, 2010*; *Chen, Wang & Wang, 2016*; *Schmidt & Mueller, 1999*). To date, only a few full-length cDNA sequences have been reported in S. furcifera. Recently, *Liang et al. (2018)* used second-generation sequencing technology to sequence the transcriptome of *S. furcifera* and reported that the unigene length was 200–400 bp, accounting for 41.50% of the total number of genes, whereas those longer than 4,000 bp only accounted for 3.37%. Furthermore, 18,416 ORFs were generated based on the unigenes. In addition, the generated ORFs were compared with those in the Nr, Swiss-Prot, GO, COG, and KEGG databases, and 18,415 transcripts were annotated. In the present study, we used the PacBio SMRT sequencing platform to obtain 100,360 SSRs and 121,395 CDSs, of which 48,873 carried complete ORFs. Unigenes longer than 4,000 bp comprised 26.4% of the total number of genes, whereas those shorter than 500 bp accounted for 8.9% (Table 4). Therefore, our results show that SMRT sequencing is a useful and effective tool for acquiring reliable full-length transcripts of *S. furcifera*. Using SMRT sequencing, 95,495 transcripts were annotated, which will assist future research and the identification of gene functions in *S. furcifera*.

## CONCLUSIONS

Through the PacBio SMRT sequencing platform, a large collection of full-length transcripts were obtained. The number and mean length of the unigenes, as well as the number of complete ORFs from SMRT sequencing were much better than those from Illumina sequencing. The obtained transcriptome data can assist further studies on gene function in S. furcifera and help clarify the interaction of S. furcifera in the ecosystem.

### Funding

This research was supported by the National Natural Science Foundation of China (No. 31860505 and 31560511), the Natural Science Foundation of Guizhou Province (No. Grant [2014]2014). The funders had no role in study design, data collection and analysis, decision to publish, or preparation of the manuscript.

### Grant Disclosures

The following grant information was disclosed by the authors:

National Natural Science Foundation of China: 31860505, 31560511.
Natural Science Foundation of Guizhou Province: [2014]2014.

## Competing Interests

The authors declare there are no competing interests.

## Author Contributions

- Jing Chen conceived and designed the experiments, performed the experiments, analyzed the data, prepared figures and/or tables, authored or reviewed drafts of the paper, and approved the final draft.
- Yaya Yu performed the experiments, analyzed the data, authored or reviewed drafts of the paper, and approved the final draft.
- Kui Kang conceived and designed the experiments, analyzed the data, prepared figures and/or tables, and approved the final draft.
- Daowei Zhang conceived and designed the experiments, prepared figures and/or tables, authored or reviewed drafts of the paper, and approved the final draft.

## Data Availability

Data is available at BioSample: SAMN12612920.

## Supplemental Information

Supplemental information for this article can be found online at http://dx.doi.org/10.7717/peerj.9320#supplemental-information.

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
