# Peer review of "SMRT sequencing of the full-length transcriptome of the white-backed planthopper Sogatella furcifera"

_PeerJ, doi:10.7717/peerj.9320_

## Round 0.1 · original submission · Major Revisions

The reviewers are generally enthusiastic about your work however they raise a number of important issues that need to be addressed. Reviewer three's suggestion of incorporating Illumina reads into your assembly is good, however I would not require it (especially if your BUSCO scores are high). But otherwise please address all reviewer suggestions in a revised manuscript.

Reviewer 1 ·

Basic reporting

On the whole, the article is clear and unambiguous with professional English used throughout. However there are some sections which require further clarification ( listed below).

The article is self-contained with relevant results to the hypothesis.

Sufficient field background/context is provided. Some statements made require literary references though.

The article has a professional article structure. The CCS and Unigene sequences have been made available to the public. Release of the subreads would be greatly appreciated.

Figures
Figure 1: Unsure if it is just me, but after multiple attempts of downloading the figure, the title and axis are not loading properly and are instead being replaced with empty boxes.
Figure 2: Legend on plot could be changed to indicate what is being presented. Also underscore should be removed from z-axis title
Figure 3: Legend missing
Figure 4: Good
Figure 5: Good
Figure 6: Legend can be more descriptive of each plot. Concerned there is too much detail being present within one figure requiring the reader to zoom in to read the font

It would be extremely beneficial if a roadmap figure was provided to the reader describing the pathway the sequences took from subreads all the way to annotated transcripts

Tables
Table 1-3: Good
Table 4: Needs description below table

Comments on specific sections of the article

Line 41-42: Description is unclear on the two ways WBP affects crops. The term ‘wind burning’ is used but I am not sure of its definition and searching it online did not return an answer. I looked up the papers referenced and neither mentioned this term, closest I found was ‘hopper burn’.
Line 45-46: This information can probably be moved closer to the beginning of the introduction.
Line 74-75: Third-generation sequencing is long read sequencing. Saying third generation sequencing is SMRT sequencing is misleading as SMRT sequencing is a proprietary method owned by Pacific Biosciences. For example, Oxford Nanopore is another third-generation sequencing method.
Line 79:82: Citation is needed for these descriptions of read lengths and potential benefits of using intact high-quality long transcripts.
Line 83: Should change SMRT to SMRT sequencing.
Line 95: What and how many inbred strains were used.
Line 117-120: Description of cDNA generation should be reworded for clarity.
Line 124-131: The description of “ANGLE” does not make sense to be included under the library construction section. Additionally, the description of “ANGLE” contains almost identical wording to the abstract of the publication which originally described it.
Line 157: What parameters were used?
Line 158: The “ANGEL” pipeline is a reimplementation of Shimizu et al 2006 “ANGLE” algorithm. “ANGEL” was coded by Pacific Bioscience’s Elizabeth Tseng https://github.com/PacificBiosciences/ANGEL
Line 164: What parameters were used?
Line 190: Citation needed for MISA.
Line 196-207: Whole section is one sentence, perhaps provide more description of each database used.
Line 253-262: How many of the generated coding sequences matched and/or did not match currently known WBP proteins?
Line 270-290: Order of the presented results should match the order of the plots in figure 6. Consider changing order of results or order of plots in the figure. COG results describing functions is hard to follow.
Line 297-298: Prior to this study, were “several” or a significant higher number of full-length transcripts unknown?
Line 300-301: Can you elaborate on how the SMRT sequencing in this study is useful for genome annotation and gene function studies.
Line 304-306: I am unsure what the authors were trying to convey in these lines.
Line 309: Comparison to previous Illumina study results should be introduced in a less abrupt manner.
Line 328-329: Before this study, was RACE the only way to obtain full-length cDNA sequences?
Line 346: Consider using a more accurate or compelling word than “lots” to described the number of transcripts obtained

Special attention should be paid to the different genomics vocabulary used in the manuscript. Several times a term is used which is vague or does not accurately describe the information being presented. Additionally, a quick grammar review should fix a few relatively benign grammar errors.

Experimental design

Goals and analysis pipeline are well described. If possible, providing the code used to run analysis pipeline will greatly increase ability to replicate experiment

Validity of the findings

Results are stated, but the final sequences and annotations are not provided to support the claims. Only the 156,138 generated unigenes are provided with no accompaning annotations.

Additional comments

Nice thorough analysis

·

Basic reporting

The manuscript is generally clear and well-written. There are no major problems with English writing and grammar, but there are a number of minor wording issues that could cause comprehension difficulties. There are also some minor problems with figure labels. I have detailed these issues below with reference to line number.

General comments: It would help to define abbreviations at their first usage in the main text, even if they have been defined in the abstract.

line 75 - the phrase "without interrupting splicing" should be removed

line 77- "assembly" should be replaced with "identification", since transcripts are not assembled with SMRT sequencing.

line 81 - Insert the word "sequencing" after SMRT

line 121 - Information about ANGLE/ANGEL seems to be in the wrong section. Also, the software used was probably ANGEL (reference the github site), which is based on ANGLE (Shimizu et al 2006.)

line 136 - insert "for" after searching

line 138 - replace "comprised" with "contained"

line 140 - replace "isotypes" with "isoforms". I believe isotypes refers to classes of antibodies.

line 177 - This sentence should be reworded as follows: The transcript sequences predicted using Plek, CNCI, and CPC tools were used to search the Pfam-A and Pfam-B databases using hmmscan.

line 192 - remove the word "strict"

line 212-213 - removing low (levels of) artificial concatemers doesn't seem to be related to identifying full-length reads. This might need to be two separate sentences.

line 218 - remove "consistent"

line 280 - Species are not homologous. This section should be rewritten as "Finally, the transcripts were compared to those in the NR database. The species with the most matching transcripts were ….."

It would also be good to comment on whether those are the species expected to be most closely related to S. furcifera.

line 289 - "several" should be replaced with "many" or "most"

line 294 - add "sequencing" after SMRT

line 319 - rewrite the beginning of this sentence to However, until recently, full-length cDNA sequences could only be obtained….

Table 2 - line 7 - replace "consistency" with "consensus"

Fig. 1 - even in the high resolution image the axis labels are not legible. The characters have all been replaced by small boxes.

Fig. 2 - I would suggest mentioning the color code in the figure legend.

Fig. 3 - The figure legend should include more than just the title.

Experimental design

The research question is clearly stated and the experiment is well thought out and clearly described for the most part.

There are a few sections in the Methods (TF detection, ANGEL, lncRNAs) where it is not clear which set of transcripts (FLNC, non-redundant, HQ?) were used as input for a particular step. This information should be added.

There are also some gaps in the analysis of the transcript data once it was obtained (see Validity of the findings section below).

Validity of the findings

The availability of full-length transcripts for S. furcifera will be a huge help in efforts to annotate the recently sequenced genome. However, I have some concerns about the analysis and interpretation of the data.

1) There is no mention of classification or mapping of the transcript sequences. When I accessed the data deposited in the NCBI Sequence Read Archive, the Taxonomy Analysis performed by NCBI indicated that approximately 12% of the transcript sequences appear to be of viral origin. Unless these transcripts originate from viral genes integrated into the S. furcifera genome, they should be removed from the S. furcifera transcriptome (although they might still be interesting in their own right). The transcripts should also be mapped against the S. furcifera genome at some point in the pipeline.

2) The analysis is also lacking an assessment of transcriptome completeness. The authors should analyze their transcriptome with a metric such as Benchmarking Universal Single-Copy Orthologs (BUSCO).

3) The number of unigenes (156,138) seems very high compared to the number of non-redundant isoforms reported in insect Iso-Seq transcriptomes of similar scope (Jia et al 2018, Zhang et al 2019). The authors describe this large number of unigenes as an improvement over previous S. furcifera transcriptomes, which might be the case if this large number was due to an increase in transcript isoforms. However, Table 4 indicates that the vast majority of the unigenes represent single isoform genes. It seems likely to me that sequence errors and/or allelic differences are preventing the collapse of the transcripts into the actual number of non-redundant isoforms. In the Iso-Seq transcriptome papers mentioned above, non-redundant isoforms were either based on the high-quality transcripts (Zhang et al 2019) or a lower stringency level was used with CD-HIT (Jia et al 2018). Mapping transcripts to the S. furcifera genome could also be helpful in identifying redundant transcripts. All downstream analysis would benefit from a more accurate count of non-redundant isoforms.

·

Basic reporting

Correct gene structure is critical in downstream biological analysis. PacBio Iso-Seq provides solution to sequencing complete gene without need of having any assembly artifacts. This manuscript provides detailed analysis of data generation and processing. Although it lacks analysis to emphasize data quality and usefulness. It also lacks comparative analysis with previously published datasets.

Experimental design

Read correction – PacBio reads have high error rate. Self-correction with Quiver might not be enough. Correction with Illumina reads is critical for getting correct open reading frames (ORFs) for protein-coding genes. Do authors have any way to measure accuracy? One way to monitor progress of correction is look at the length of ORFs or CDS.

Another issue with Iso-Seq is the depth of sequencing. There is chance of missing genes based on coverage along with diversity of transcripts in the stage/tissues used for sequencing. To generate comprehensive transcriptome, combining Illumina RNA-Seq reads based transcripts could lead to more complete representation of the transcripts. To be useful for broader community, authors should integrate public RNA-Seq datasets with Iso-Seq to generate comprehensive transcriptome.

Methods - Library preparation
It mentions about CDS predictions using ANGLE. Seems redundant and out of place as this is also discussed later in the CDS prediction section.

Validity of the findings

lncRNA are important class of non-coding genes. lncRNA are spliced genes and numbers of exons is critical parameters in classifying lncRNA. I would recommend to show mapping statistics to the genome for lncRNA along with the rest of the Iso-Seq transcripts.

Do authors have BUSCO single copy marker statistics for Iso-Seq based transcriptome?

---

## Round 0.2 · Minor Revisions

Thank you for addressing the majority of the reviewers comments. There are still a few remaining issues as outlined in the current reviews. Please thoroughly address these points in a revised manuscript.

Reviewer 1 ·

Basic reporting

The article is clear and unambiguous with professional English used throughout.
There still remains some sections which require further clarification (listed below).
The article is self-contained with relevant results to the hypothesis.
Sufficient field background/context is provided.
The article has a professional article structure.
The 630,447 circular consensus sequences can be found at https://www.ncbi.nlm.nih.gov/biosample/SAMN12612920/

Line 74-77 – Reorganize sentence to: Single-molecule real-time (SMRT) sequencing, a third-generation sequencing technology developed by Pacific Biosciences (PacBio), can effectively overcome this problem (Eid et al. 2009; Korlach et al. 2010).
Line 147-149 – Clarify that insecta_odb10 was the dataset used in BUSCO analysis. Should it be transcriptome mode instead of protein mode? Was BUSCO analysis done on the full-length reads, FLNCs, or polished consensus isoforms? This sentence also should be moved to a different location, currently interrupts the flow of the sentences surrounding it.
Line 166-168 – Reorganize sentence to : The ANGEL pipeline, which is based on ANGLE (Shimizu et al. 2006), was used to identify the protein coding regions from cDNAs (https://github.com/PacificBiosciences/ANGEL).
Line 202-203 – Reword sentence to: The unit sizes and their minimum number of repetitions were: 1-10, 2-6, 3-5, 4-5, 5-5, and 6-5
Line 232-233 – This part should be placed next to the description of the reads used in the analysis. It should also be reworded for grammar.
Line 293 – Remove “were”
Line 298 – Change “was” to “is”
Line 299 – Uncapitalize “The”
Line 332 – Change “readings” to “reads” or “transcripts”
Line 334-335 – Change to: The lengths of the FLNC transcripts were aligned to the sizes of the transcriptome library.
Line 332-335 – Confusing switches between singular and plural objects
Line 335-336 – Need to reword for clarity
Line 342 – Change “from Illumina sequencing result” to “assembled using only Illumina sequencing”.
Line 361-364 – Change to “However, most methods used to obtain a full-length cDNA clone are time consuming, expensive, and inefficient (Bower & Johnston 2010; Chen et al. 2016; Schmidt & Mueller 1999).
Line 380-381 – Change to “Through the PacBio SMRT sequencing platform, a large collection of full-length transcripts were obtained.

Experimental design

Goals and analysis pipeline are well described. If possible, providing the code used to run analysis pipeline will greatly increase ability to replicate experiment.

Validity of the findings

Results are stated. The 156,138 generated unigenes are provided with no accompaning annotations.

·

Basic reporting

All of my original concerns have been addressed, but a few minor errors have been introduced during the editing process.

lines 125-126. The following lines seem to be out of place. They may be left from the ANGLE material, most of which was deleted. "During the CDS prediction process, we kept the best sequence as the final result. If there were multiple sequences that could not be filtered, all of them were output."

lines 158-159. The phrase "which is based on ANGLE (Shimizu et al. 2006)" should be placed immediately after "The ANGEL pipeline".

line 326-327 - Reword to " However, until recently, the methods for obtaining full-length cDNA clones were time consuming, expensive, and inefficient."

line 344 - Replace "a plenty of" with either the word "numerous" or an exact number.

line 345 - Add "as well as" after the comma.

Experimental design

The one issue I still have with the experimental design is the lack of mapping of transcripts to the S. furcifera genome. The authors responded that they do not have access to the genome, but I was able to download the assembly described by Wang et al (2017) from http://gigadb.org/dataset/view/id/100255/File_page/2. The BUSCO statistics for the Wang et al genome indicate a high degree of completeness, so a comparison should be very informative.

Validity of the findings

Interpretation of the current findings appears to be valid.

---

## Round 0.3 · Minor Revisions

There were two reviewer comments that you did not respond to. Can you please comment on the following two points. If you are not able to do these, please explain.

Reviewer 1: "If possible, providing the code used to run analysis pipeline will greatly increase ability to replicate experiment."

Reviwer 2:
"The one issue I still have with the experimental design is the lack of mapping of transcripts to the S. furcifera genome. The authors responded that they do not have access to the genome, but I was able to download the assembly described by Wang et al (2017) from http://gigadb.org/dataset/view/id/100255/File_page/2. The BUSCO statistics for the Wang et al genome indicate a high degree of completeness, so a comparison should be very informative."

---

## Round 0.4 · accepted · Accept

Thank you for adding the analysis of mapping your reads to the published genome reference. I am happy to recommend this article for publication.